# Assessment of Heavy Metal(oid)s Accumulation in Eggplant and Soil under Different Irrigation Systems

**Yasir Tariq** [1], **Nusrat Ehsan** [2,*], **Umair Riaz** [3,*], **Rabiya Nasir** [2], **Waqas Ahmed Khan** [4], **Rashid Iqbal** [5], **Shehzad Ali** [6], **Eman A. Mahmoud** [7], **Izhar Ullah** [8] and **Hosam O. Elansary** [9]

1   Department of Environmental Sciences, University of the Lahore, Lahore 5400, Pakistan; yasir.tariq@hmail.com
2   Department of Environmental Management, National College of Business Administration and Economics, Lahore 54660, Pakistan
3   Department of Soil & Environmental Sciences, MNS-University of Agriculture, Multan 60000, Pakistan
4   Sustainable Development Study Center, Government College University, Lahore 54000, Pakistan
5   Department of Agronomy, Faculty of Agriculture and Environment, The Islamia University of Bahawalpur, Bahawalpur 63100, Pakistan
6   Department of Environmental Sciences, Quaid-i-Azam University, Islamabad 45320, Pakistan
7   Department of Food Industries, Faculty of Agriculture, Damietta University, Damietta 34511, Egypt
8   School of Biomedical Sciences, Queensland University of Technology, Brisbane City 4000, Australia
9   Department of Plant Production, College of Food & Agriculture Sciences, King Saud University, P.O. Box 2460, Riyadh 11451, Saudi Arabia
*   Correspondence: nusrat.ncbae@gmail.com (N.E.); umair.riaz@mnsuam.edu.pk (U.R.); Tel.: +92-30-0620-8789 (U.R.)

**Abstract:** Heavy metal(oid)s (HMs) contamination in soil directly related to food contamination and human health. This study was conducted to investigate the effect of HMs accumulation in eggplant irrigated through different water sources. Water samples were collected from three distinct sources, namely urban and rural sewage, urban and rural canal water, and urban and rural tube well water. A total of 20, 9, and 6 samples were obtained from each respective source. Soil samples were collected, with three replications each, from two layers i.e., 0–15 cm and 15–30 cm depth. Results depicted that, in irrigation water samples, turbidity was in this order: sewage water > canal water > tube well samples, while average total dissolved solids TDS) was in this order: canal water > sewage water > and tube well water. The overall HMs concentrations in all three sources of irrigation water samples were above the permissible limits of Punjab Environmental Quality Standards PEQs). The level of HMs in vegetables varies by the ability of plants to selectively translocate from the contaminated soil to the plant's fruit (eggplant). Overall, the translocation factor of HMs in plant samples was in this order: Cadmium (Cd) > Zinc (Zn) > Nickel (Ni) > Iron (Fe) > Lead (Pb). On the basis of this study, it is concluded that irrigation source, especially sewage, is a major source of heavy metal contamination in eggplant. The study highlights the need for further research on gaps in the implementation of policies for irrigation water quality.

**Keywords:** contamination; sewage; heavy metals; accumulation; health; toxic; groundwater

## 1. Introduction

The HMs contamination of soil and agricultural products is a global concern, as it accumulates in plant and transfers to the primary and secondary consumers, including humans [1,2]. The concentration and absorption of HMs in soil depends upon the physio-chemical characteristics of soil, such as pH; as the pH is lower, the HM solubility increases, hence causing higher accumulation in plants and greater human health risks [3,4]. The irrigation of agricultural fields with wastewater has been common practice for decades, which causes the accumulation of excessive HMs in crops and other agricultural products [5,6]. Municipal and industrial wastewater usage for irrigation is also a common practice in

developing countries, which have a high potential to cause HM accumulation of soil and, hence, bio-accumulation in the food chain [7]. In addition to the irrigation from wastewater, the HMs may enter the agricultural fields through atmospheric depositions, dumping of waste, and use of synthetic fertilizers, pesticides [8], and other chemicals [9]. The chemical and structural characteristics of soil, such as soil type and moisture content, including organic carbon, cation exchange capacity, and soil mineral content, also play a significant role in HMs accumulation of soil [10]. The major HMs of global concern for human health include Chromium (Cr), Nickel (Ni), Lead (Pb), Arsenic (As), and Iron (Fe) [11]. Sewage and wastewater irrigation have been identified as potential sources of HMs, including Cr, Cd, Ni, Copper (Cu), Pb, and Zn in food items [12].

Pakistan is facing water scarcity, as the current resources are insufficient to satisfy irrigation needs, as well as other water consumption needs, due to an increase in population and changes in the consumption pattern [13]. Demand for water, particularly for irrigation purposes, is increasing due to increasing populations, while the increase in urbanization results in ever-increasing wastewater quantities. The treatment of wastewater is very expensive due to the higher import cost of such chemicals in developing countries. In many areas of developing nations, including Pakistan, the sewage water is used for irrigation purposes directly or after preliminary treatment. It includes significant quantities of toxic HMs that cause issues [14]. Pakistan is basically an agricultural country, but most of its agriculturally productive area falls in the arid and semi-arid climate. Around one-third of the uses of groundwater are for drinking, agricultural purpose, and the industrial sector.

Urban and peri-urban farming depends, at least to certain degree, on the sewage water as somewhat of a water irrigation source in the urban regions of many (developing) countries. In different plants, HMs can bring high-level toxicity by: breaking the strong bonding of vital bio-molecules, changing the active sites of specific enzymes, producing Reactive Oxidative Species (ROS) which change the antioxidant defense systems, changing several macromolecules, and replacing fundamental metal ions within the structural formula of bio-molecules [15–17].

Untreated sewage waste, canal water, as well as ground water are extensively used for irrigation in Punjab; however, their accumulation in soil and plants, as well as their impacts on human health are not studied much. Long-term utilization of municipal or industrial wastewater in irrigation may result in HMs being accumulated in agricultural soils and crops [7].

The accumulation of HMs in crops and vegetables is associated with many diseases [5,6,11]. Heavy metals, such as Cd, in crops causes toxicity in human beings and livestock. In renal tubular osteomalacia or itai-itai disease, Cd is the main etiopathogenetic factor. It is well recognized that the rice crop has a greater capability to accumulate Cd than other cereal crops [16]. In children, it may cause reduced intelligence, learning disabilities, impaired development, coordination problems, cardiovascular diseases, renal failure, and short-term memory loss problems [18,19]. In developing and under-developed countries, the raw sewage water is used for irrigation [20]. Long-term human intake of HM-contaminated food, such as cereals, vegetables, and others, may cause severe impacts on the biochemical and biological systems of humans. Plants—in particular, leafy plants—cultivated in the HM-contaminated soils accumulate more than metals cultivated in uncontaminated soils because they absorb the metals in their leaves [21]. Most vegetables are leafy and cultivated in polluted soil, leading to a larger buildup of HMs in their leaves [22]. The HM contents (Pb and Cd) were found higher in eggplant samples as compared to the permissible limits [23]. Crop irrigation with wastewater has a negative impact and has a serious risk to health due to the consumption of such wastewater-polluted food crops [24]. The food crops' edible components, highly contaminated with poisonous metals, have a negative effect on users [25–27]. Cd is considered poisonous, and its prolonged exposure to reduced concentrations leads to an accumulation in the kidneys and, probably, delicate bones, and kidney and lung damage, hypertension, anemia, arthritis, diabetes, cardiovascular disease, cancer, cirrhosis, decreased fertility, osteoporosis, hypoglycemia, renal illness, headache, and stroke are some strange long-term outcomes [28].

Cr$^{+6}$ is a renowned toxin and human carcinogenic metal [29]. Other types of effects include abdominal pain, arthritis, and anemia, a deficit of care, blindness, back tissues, cancer, convulsions, constipation, depressions, diabetes, headaches, tooth decay, migraine, and thyroid imbalances [29]. Crops' exposure to these HMs have harmful consequences for human health, and their usage causes brain, bone, kidney, and heart disease. In addition, Cd and Mercury (Hg) can cause cancer [30–32]. Health effects of exposure to such HM accumulation in soil, water bodies, and their pattern of translocation and bioaccumulation in plants and humans need to be assessed and quantified [13,30,33–35]. The translocation of HMs in plants is a major source of bioaccumulation in the human body. The present study has been conducted to measure the concentration of HMs in different types of irrigation water sources, as well as its accumulation in soil and from soil into the eggplant. Moreover, the translocation factor of HMs from soil to the eggplant is also calculated.

## 2. Materials and Methods

The schematic diagram of the present research is presented in Figure 1. There are three main irrigation sources, i.e., sewage, canal, and ground water (extracted from tube wells). Irrigation water samples were collected from these three sources at both urban and rural agricultural lands. Soil and vegetable samples were also collected for each irrigation source. These water, soil, and vegetable samples were analyzed for selective physico-chemical parameters and heavy metal(oid) concentrations. On the basis of these results, the translocation factor (TF) was calculated as the formula given below:

$$\text{Translocation Factor (TF)} = \frac{\text{Concentration of heavy metals in vegetative parts}}{\text{Concentration of heavy metals in roots}}$$

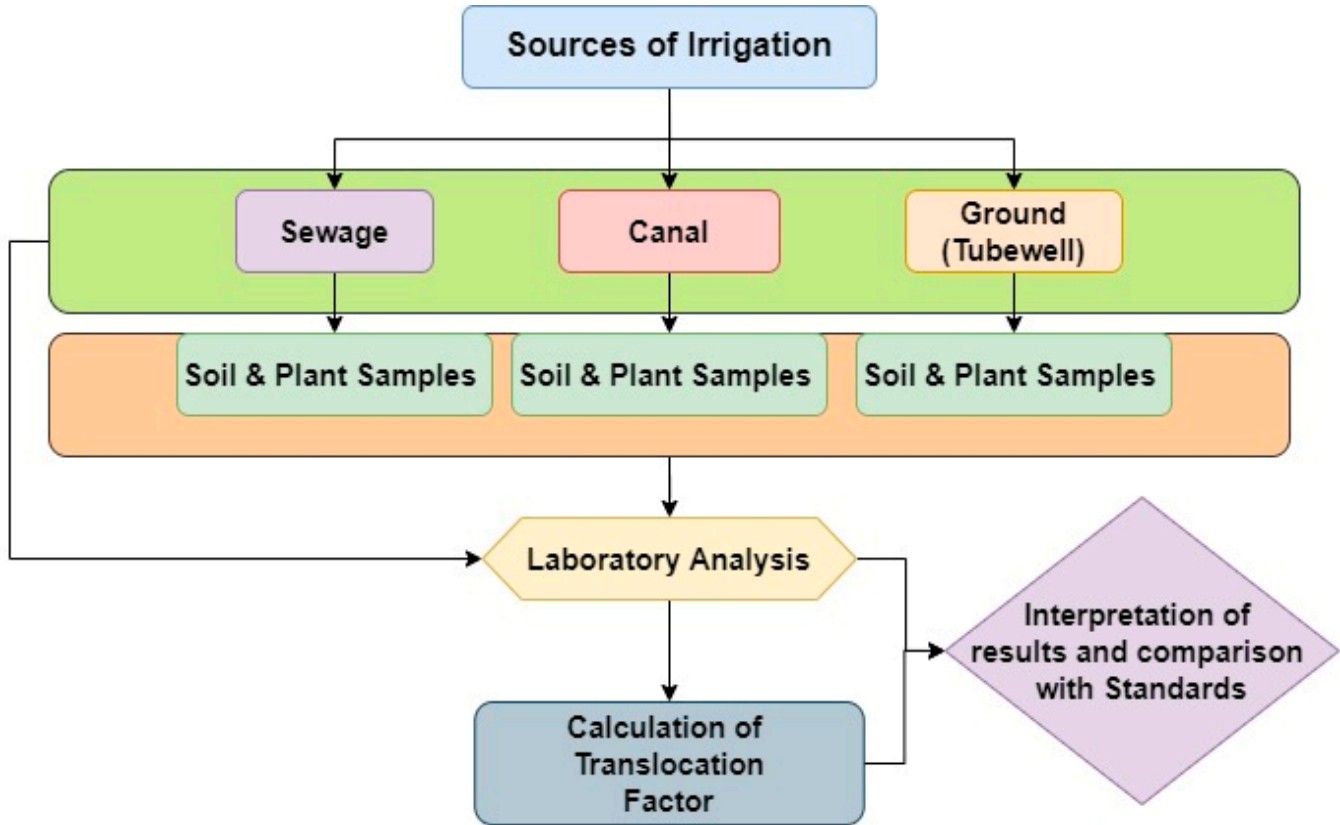

**Figure 1.** Schematic diagram of research.

While the TF results were compared with standards, the schematic diagram for the experimental design of each phase is shown below.

### 2.1. Area of Study

This research was carried out in both urban and rural areas of the Gujranwala district of Punjab, Pakistan. Punjab is the agricultural hub of Pakistan, with a wide variety of crops and vegetables which are exported to the other regions of the country. Gujranwala is a large town in eastern Pakistan, and it lies between latitude $32°9'58.8636''$ N and $74°11'45.2400''$ E. The city has a population of about 1,384,471 people with a major population serving in the manufacturing and agricultural sectors of the country [36]. There are many small-to-medium-sized industrial units of leather, food products, metal and surgical equipment, and textiles. This region falls in the hot semi-arid climatic zone, with a maximum temperature of 36–42 °C in summer (June to September) and a minimum of 5 °C during winter (November to February). The highest precipitation occurs during monsoon (July and August), i.e., the average annual precipitation is around 24 mm [36].

### 2.2. Sample Collection, Preparation, and Analysis

The soil, water, and vegetable sample collection were conducted in September 2021. At this time of the year, the eggplants are mature and ready for harvest. All the procedures for collection, preservation, and transportation were according to the standard methodologies of American Public Health Association (APHA) (1999) [29,37].

Irrigation in this region is conducted through tube well, canal (Hardopur village on Hafizabad Road), and sewage water (drains which contains municipal as well as industrial effluent, which is emitted after primary treatment). Water samples from these three sources were collected in three replications. The number of water samples collected from urban sewage water (SW1) and rural sewage water (SW2) were 20 each, followed by urban canal water (CW1) and rural canal water (CW2) with 9 samples each, and urban tube well water (TW1) and rural tube well water (TW2) with 6 samples each. Pre-sterilized polyethylene airtight plastic cap bottles (150 mL each), after immediate acidification with 1 mL nitric acid, were used to avoid surface assimilation of heavy metal(oid)s with the bottles walls and to keep the metals in solution [29,37].

Soil samples were collected through stratified random samplings with three replications each. The soil samples were collected from the top two layers, i.e., 0–15 cm and 15–30 cm depth by digging a monolith of 10 × 10 × 15 cm. The soil samples were air dried under shade and sieved (sieve size = 2 mm). These samples were stored at room temperature until chemical analysis was conducted.

Eggplant (*Solanum melongena*) samples were collected (about 2 kg) from each irrigation source, i.e., canal water and tube well-irrigated vegetable samples from Hardopur fields on Hafizabad Road, as well as sewage water-irrigated vegetables from Khayali Gujranwala and tube well water from field of Gujranwala. The samples were carefully labeled, washed to remove dust and other pollutants, air dried (for six days), and transported to the laboratory in polythene bags. Samples were dried at 120 °C for 8 h in an electric oven to eliminate moisture. The dried samples were ground into powder manually with a mortar and pestle and, then, sieved (mesh size = 2 mm). The specimens were finally taken in polyethylene bags and kept in desiccators until they were digested [38].

The plants were separated (fruit only) to find out the translocation factor. The plant samples were washed with tap water and rinsed twice with distilled water; then, they were dried at 80 °C for 24 h. After drying, they were grounded (pulverized with a micro hammer mill), and then, 2.0 g of sample powders was weighed into 100 mL Pyrex beakers and treated with 10 mL concentrated $HNO_3$. The beaker was covered with watch glass, and the suspension was heated to 130 °C for 1 h. A total of 30 mL of HCl was added [38].

The soil, water, and vegetable samples were digested (di-acid method) using the standard methodology of International Center for Agricultural Research in the Dry Areas (ICARDA) and American Public Health Association (APHA) and tested using Atomic absorption Spectroscopy (Aurora AI-1200). The linearity of the adopted method was $R^2 > 0.99$, accuracy was 99.9%, and limit of detection (LOD) was 0.001 mg kg$^{-1}$, while limit of quantification (LOQ) was 0.01 mg kg$^{-1}$. The comparative analysis was performed

with the Punjab Environmental Quality Standards (PEQs), while vegetable results were compared with World Health Organization (WHO) and FAO. The permissible limits of studied heavy metal(oid)s are mentioned in Table 1.

**Table 1.** HMs and their Permissible limits.

| HMs | Permissible Limit in Soil (EU) mg kg$^{-1}$ | Permissible Limit in Eggplant Vegetable (WHO/FAO) mg kg$^{-1}$ | Permissible Limit in Canal, Ground & Wastewater (PEQS) mg L$^{-1}$ |
|---|---|---|---|
| Zinc (Zn) | 300 | 40 | 5.00 |
| Iron (Fe) | 5000 | 425.5 | 8.00 |
| Cadmium (Cd) | 3 | 0.05–0.2 | 0.10 |
| Chromium (Cr) | 150 | 2.3 | 1.00 |
| Nickel (Ni) | 75 | 1.5 | 0.01 |
| Lead (Pb) | 300 | 0.05–0.3 | 0.50 |

*2.3. Statistical Analysis*

The obtained data were analyzed statistically by using computer based software, i.e., OriginLab 2021. The descriptive statistics were also applied.

### 3. Results

The physical characteristics indicated that sewage water samples turbidity was above the permissible limits of PEQs (5 NTU), i.e., SW1 (8.80 ± 1.26 NTU) and SW2 (8.50 ± 1.33 NTU). In canal water samples, the turbidity was within the permissible range of PEQs, i.e., CW1 (3.45 ± 2.01 NTU) and CW2 (4.50 ± 1.34 NTU). The lowest turbidity was observed in tube well samples, i.e., TW1 (2.54 ± 1.48 NTU) and TW2 (2.76 ± 1.03 NTU). The pH in irrigation water samples was highest in sewage water, followed by canal water, and least for the tube well water. It was observed that pH in tube well water is within the permissible range of PEQs, i.e., 6.5–7.5, while it was higher for sewage and canal water samples. The average total dissolved solids in canal water were highest, i.e., 986 and 990 mg L$^{-1}$, followed by sewage water TDS, i.e., SW1 (800 mg L$^{-1}$) and SW2 (708 mg L$^{-1}$) and TW1 (817 mg L$^{-1}$) and TW2 (686.4 mg L$^{-1}$). The graphical representation of physcio-chemical parameters is provided in Figure 2. The correlation of physio-chemical characteristics of irrigation water is mentioned in Table 2.

**Table 2.** Pearson Correlation between physio-chemical parameters of irrigation water.

| | pH | TDS | Turbidity | Alkalinity | Cr | Fe | Zn | Cu | Mg | Fe | Ni |
|---|---|---|---|---|---|---|---|---|---|---|---|
| TDS | 0.058 | | | | | | | | | | |
| | 0.767 | | | | | | | | | | |
| Turbidity | 0.33 | 0.495 | | | | | | | | | |
| | 0.08 | 0.006 | | | | | | | | | |
| Alkalinity | 0.039 | 0.466 | 0.474 | | | | | | | | |
| | 0.839 | 0.011 | 0.009 | | | | | | | | |
| Cr | 0.215 | 0.168 | 0.002 | 0.097 | | | | | | | |
| | 0.262 | 0.384 | 0.992 | 0.616 | | | | | | | |
| Fe | 0.278 | 0.044 | −0.087 | −0.187 | 0.406 | | | | | | |
| | 0.144 | 0.822 | 0.655 | 0.333 | 0.029 | | | | | | |
| Zn | 0.11 | 0.475 | 0.444 | 0.417 | −0.054 | 0.185 | | | | | |
| | 0.571 | 0.009 | 0.016 | 0.024 | 0.779 | 0.337 | | | | | |

**Table 2.** *Cont.*

|  | pH | TDS | Turbidity | Alkalinity | Cr | Fe | Zn | Cu | Mg | Fe | Ni |
|---|---|---|---|---|---|---|---|---|---|---|---|
| Cu | −0.09 | −0.443 | −0.02 | −0.197 | −0.098 | −0.18 | −0.089 |  |  |  |  |
|  | 0.643 | 0.016 | 0.92 | 0.305 | 0.611 | 0.351 | 0.647 |  |  |  |  |
| Mg | −0.009 | −0.299 | −0.245 | −0.235 | 0.306 | 0.172 | −0.041 | 0.33 |  |  |  |
|  | 0.962 | 0.115 | 0.2 | 0.219 | 0.106 | 0.372 | 0.831 | 0.08 |  |  |  |
| Fe | −0.114 | −0.016 | −0.333 | 0.317 | 0.219 | −0.245 | −0.146 | −0.038 | 0.097 |  |  |
|  | 0.556 | 0.933 | 0.077 | 0.094 | 0.255 | 0.201 | 0.45 | 0.846 | 0.618 |  |  |
| Ni | −0.151 | −0.39 | −0.036 | −0.217 | −0.115 | −0.091 | −0.083 | 0.179 | 0.063 | −0.122 |  |
|  | 0.433 | 0.839 | 0.853 | 0.258 | 0.554 | 0.639 | 0.668 | 0.352 | 0.744 | 0.529 |  |
| Cd | 0.044 | 0.132 | 0.214 | −0.032 | 0.14 | 0.436 | −0.174 | −0.17 | −0.13 | −0.036 | −0.036 |
|  | 0.819 | 0.494 | 0.265 | 0.243 | 0.867 | 0.469 | 0.018 | 0.336 | 0.378 | 0.5 | 0.853 |

Note: Table content: Correlation coefficient; *p* value < 0.05 indicate significant correlation.

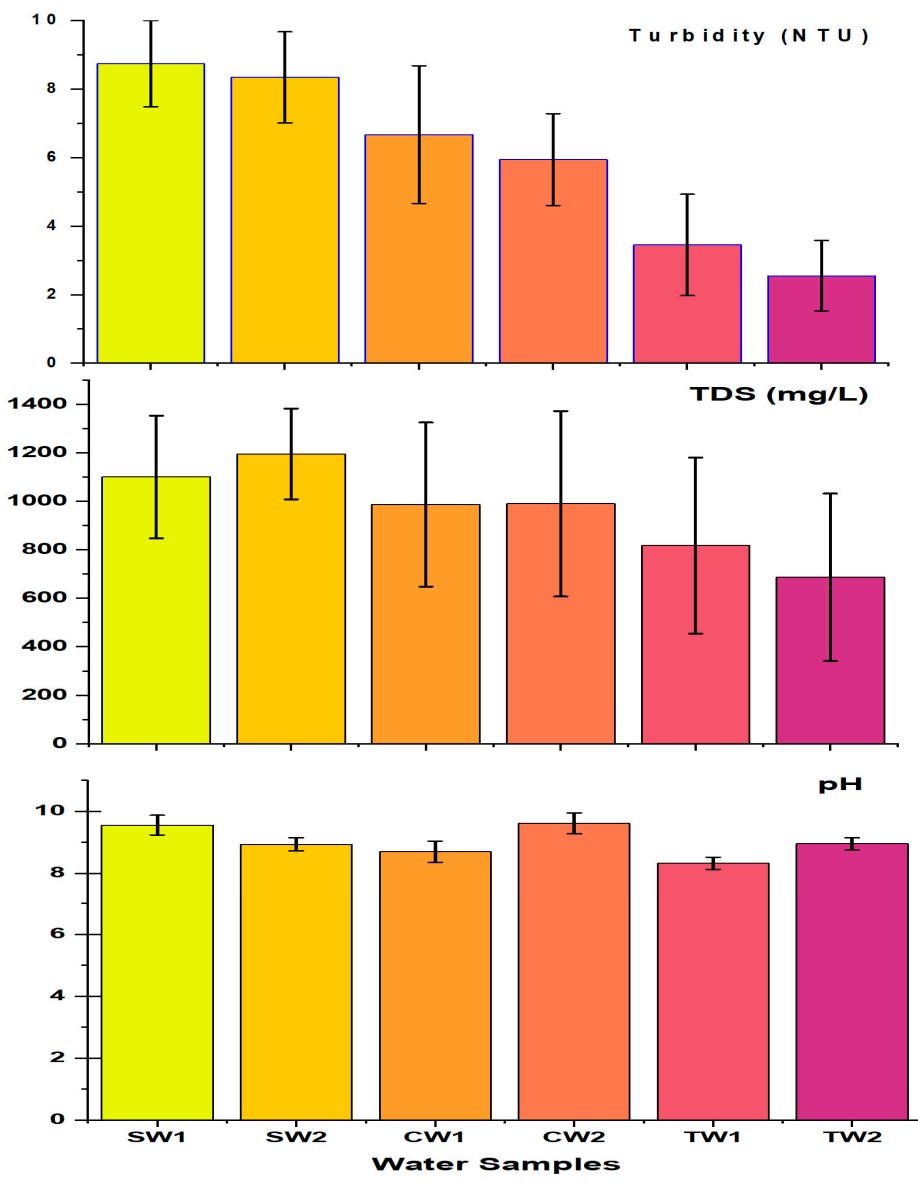

**Figure 2.** Characteristics of irrigation water samples.

### 3.1. Level of HMs in Wastewater Samples

The overall heavy metal(oid) concentrations in all three sources of irrigation water samples were above the permissible limits of PEQs (Figure 3). The HMs level in sewage water was highest with concentrations in this order: Fe ($9.1 \pm 0.4$ mg L$^{-1}$), Zn ($7.5 \pm 0.08$ mg L$^{-1}$), Ni ($3.9 \pm 0.5$ mg L$^{-1}$), Pb ($3.5 \pm 0.36$ mg L$^{-1}$), Cd ($2.6 \pm 0.26$ mg L$^{-1}$), and Cr ($2.3 \pm 0.2$ mg L$^{-1}$). The results also indicated that the highest concentration of HMs (except Ni) was found in the order of sewage water > canal water > tube well samples. Moreover, it was also observed that Ni was only present in the sewage water samples and not in tube well and canal water. The levels of HMs in canal water samples were, in order, Fe ($8.5 \pm 0.15$ mg L$^{-1}$) > Zn ($6.1 \pm 0.25$ mg L$^{-1}$) > Cr ($1.7 \pm 0.25$ mg L$^{-1}$) > Ni ($1.7 \pm 0.25$ mg L$^{-1}$) > Cd ($1.4 \pm 0.25$ mg L$^{-1}$ > Pb ($1.2 \pm 0.35$ mg L$^{-1}$). The level of Fe in canal water samples ranged from $8.2 \pm 0.15$–$8.5 \pm 0.15$ mg L$^{-1}$. All of the samples (100%) were above the permissible limits, i.e., 8 mg L$^{-1}$. The level of Cr in canal water samples ranged from $1.1 \pm 0.20$–$1.7 \pm 0.25$ mg L$^{-1}$. All of the samples (100%) were above the permissible limits, i.e., 1 mg L$^{-1}$. The Ni concentration in canal water ranged from $0.9 \pm 0.3$ to $1.7 \pm 0.25$ mg L$^{-1}$. All of the samples (100%) exceeded the allowable value of WHO, i.e., 0.01 mg L$^{-1}$, and PEQS, i.e., 1 mg L$^{-1}$. The level of Cd in canal water samples ranged from $0.8 \pm 0.20$ to $1.4 \pm 0.25$ mg L$^{-1}$. All of the samples (100%) were above the permissible limits, i.e., 0.1 mg L$^{-1}$. The level of Pb in canal water samples ranged from $0.5 \pm 0.26$ to $1.2 \pm 0.45$ mg L$^{-1}$. All of the samples (100%) were above the permissible limits, i.e., 0.5 mg L$^{-1}$.

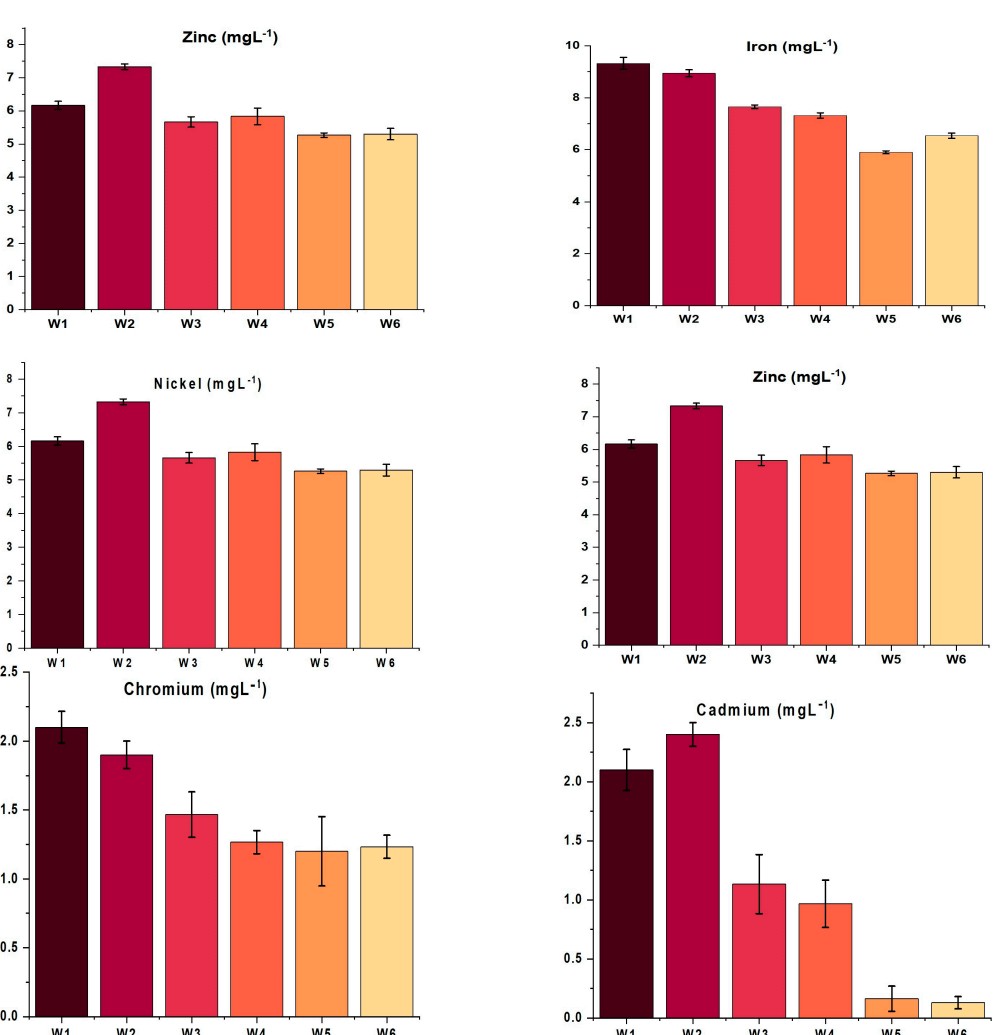

**Figure 3.** Heavy metal(oid) concentration in irrigation water samples in the study area.

### 3.2. Heavy Metal(oid) Level in Soil Samples

The findings of heavy metal(oid) accumulations in soil (Figure 4) showed that Zn, Fe, Pb, Cr, Ni, and Cu showed relatively greater values for wastewater, canal, and ground water-irrigated lands. The HM level in wastewater soil samples were in this order: Fe ($18,154 \pm 42.25$ mg L$^{-1}$) > Pb ($348.9 \pm 0.70$ mg L$^{-1}$) > Zn ($313.2 \pm 0.35$ mg L$^{-1}$) > Cr ($179.4 \pm 0.45$ mg L$^{-1}$) > Ni ($86.4 \pm 0.68$ mg L$^{-1}$) > Cd ($7.6 \pm 0.17$ mg L$^{-1}$). This shows that the wastewater has more sources of these metals than tube well and canal water.

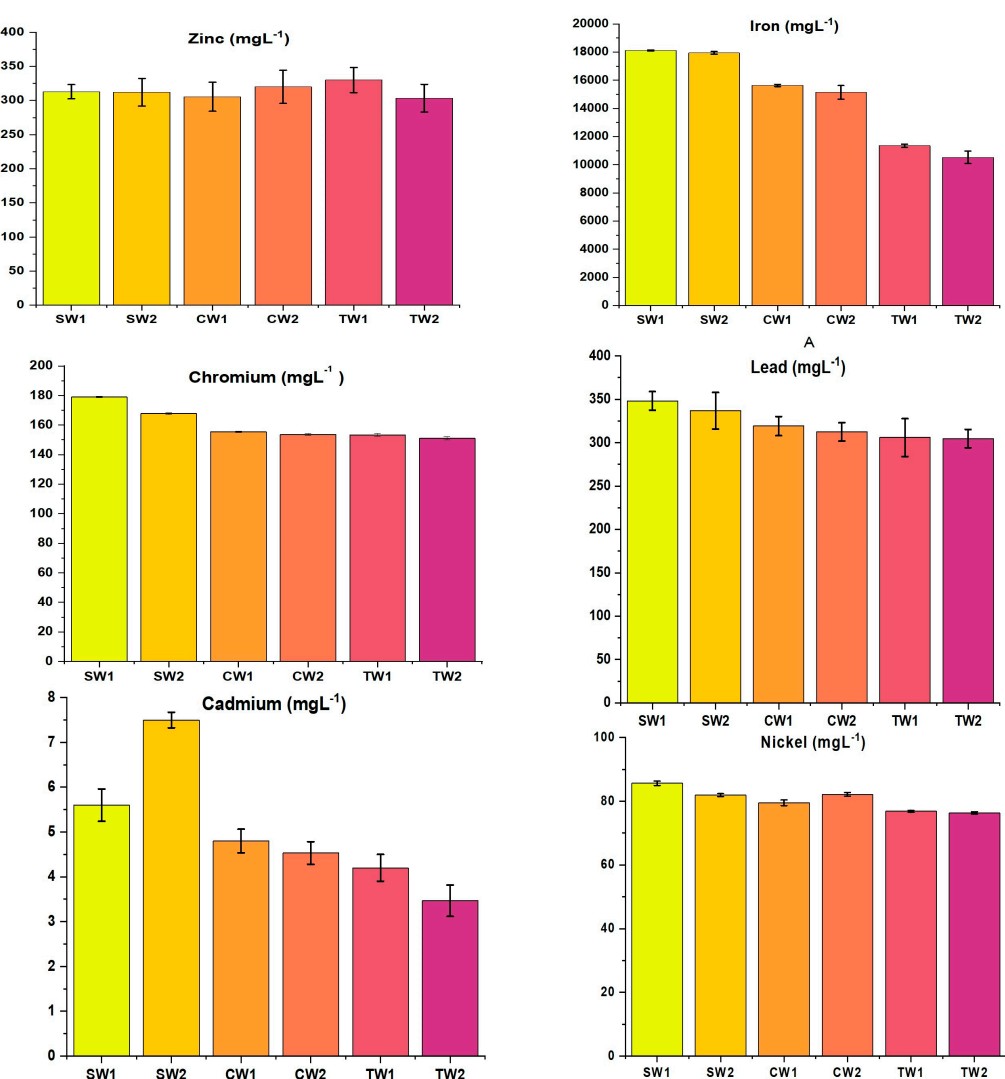

**Figure 4.** Comparison of heavy metal(oid) concentrations in soil samples irrigated with sewage water, canal water, and tube well water.

### 3.3. Heavy Metal(oid) Concentration in Vegetable Samples

The level of HMs in vegetables varies by the ability of plants to selectively translocate from the contaminated soil to the plant's fruit (eggplant) and accumulate some of these elements (Figure 5). The concentration of Zn in eggplant samples ranged from $41.2 \pm 0.15$ mg kg$^{-1}$ (tube well-irrigated sample) to 50.6 mg kg$^{-1}$ (sewage-irrigated sample). The Fe concentration varied from $426.8 \pm 0.35$ mg kg$^{-1}$ to $433.4 \pm 0.40$ mg kg$^{-1}$. The concentration of Cd ranged from $4.5 \pm 0.36$ to $0.86 \pm 0.15$ mg kg$^{-1}$. The Cr concentration in eggplant was highest in samples irrigated with sewage water, i.e., $4.8 \pm 0.17$ mg kg$^{-1}$, followed by canal water $3.2 \pm 0.65$ mg kg$^{-1}$ and tube well water $3.03 \pm 0.25$ mg kg$^{-1}$, respectively. Similarly, the Ni in plant samples was observed ranging from $1.7 \pm 0.2$ to $4.4 \pm 0.36$ mg kg$^{-1}$ in tube well water and sewage water, respectively. Previous studies

show that exposure to the large levels of Cr causes skin irritation, circulatory tissue harms outcomes, ulceration, and nervous tissue, causing health problems.

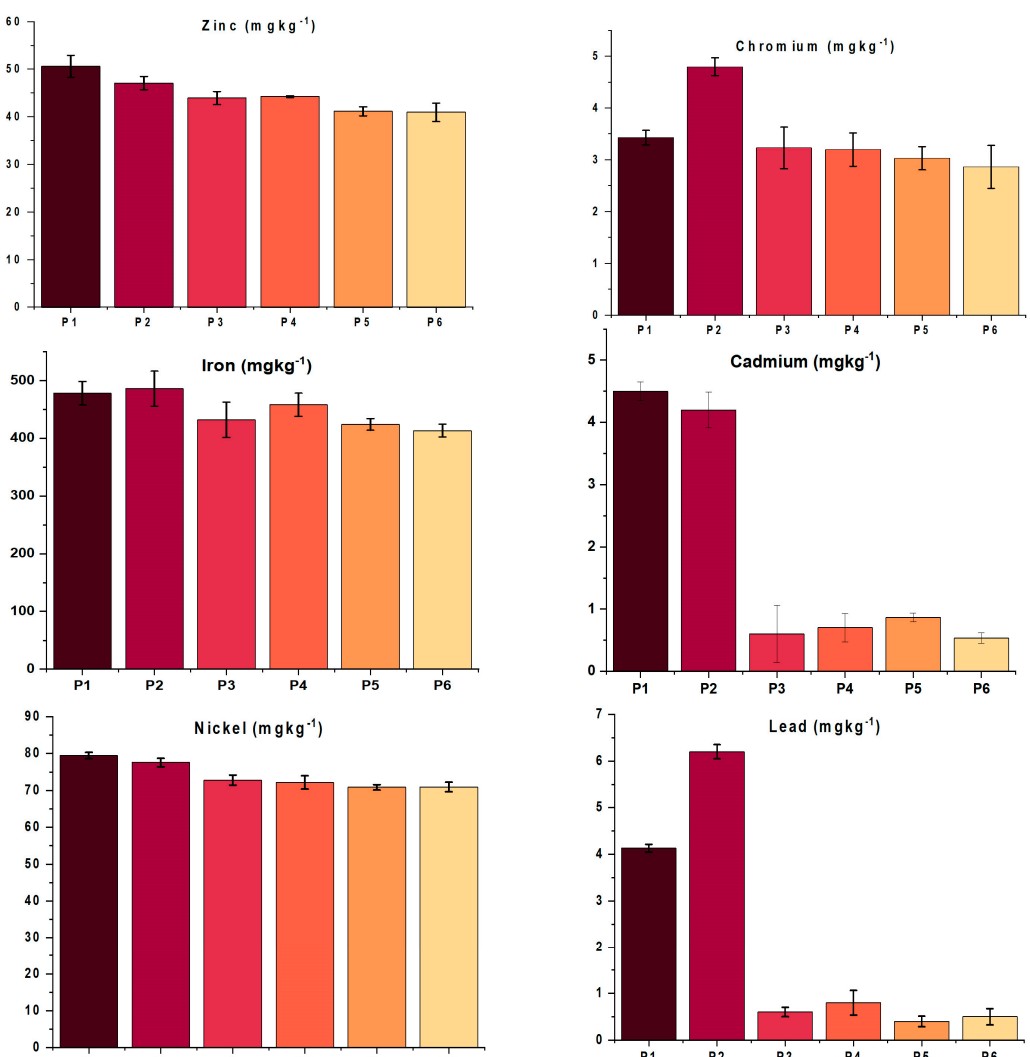

**Figure 5.** Comparison of heavy metal(oid) concentration in plant samples irrigated with sewage water, canal water, and tube well water.

*3.4. Translocation Factor*

The translocation factor of heavy metal(oid)s in plants was observed in the order of Cd > Zn > Ni > Fe > Cr > Pb (Table 3).

**Table 3.** Translocation factor of heavy metal(oid)s from soil to the plant.

|  | Zn | Fe | Cd | Cr | Ni | Pb |
|---|---|---|---|---|---|---|
| TF1 | 0.15 | 0.02 | 0.80 ± 0.01 | 0.02 ± 0.00 | 0.07 | 0.01 |
| TF2 | 0.15 | 0.02 | 0.56 ± 0.06 | 0.03 ± 0.00 | 0.11 | 0.02 |
| TF3 | 0.14 | 0.00 | 0.12 ± 0.00 | 0.02 ± 0.00 | 0.01 | 0.00 |
| TF4 | 0.14 | 0.00 | 0.15 ± 0.08 | 0.02 ± 0.00 | 0.02 | 0.00 |
| TF5 | 0.14 | 0.01 | 0.20 ± 0.05 | 0.02 ± 0.01 | 0.01 | 0.00 |
| TF6 | 0.14 | 0.01 | 0.15 ± 0.06 | 0.06 ± 3.40 | 0.03 | 0.00 |

## 4. Discussion

The results of the present study shows the rate of heavy metal(oid) accumulation, from different water sources, into the soil and its uptake by eggplant. Significant uptake and accumulation in eggplant is observed for heavy metal(oid)s such as Cr, Ni, Cd, and Pb. These findings are in-line with the findings of Hamid et al. [26] and Ismail et al. [39], as they showed accumulation of Cr, Ni, and Pb in plant samples irrigated through canal water.

The Fe concentrations found in canal water samples by [39] were 1300 mg $L^{-1}$, which are higher than our findings. The level of Zn in canal water samples ranged from $5.5 \pm 0.20$ to $6.1 \pm 0.25$ mg $L^{-1}$. All of the samples (100%) were greater than the allowable limits, i.e., 5 mg $L^{-1}$. Ismail et al. [39] studied and found the highest level of Zn, i.e., 650 mg $L^{-1}$ in canal water, which supports the finding of this study. The findings of Khan et al. [33] support the findings of this research, as the level HMs were high as compared with the allowable values in the order of Fe > Ni > Zn > Cd > Cr. The Fe concentrations found in wastewater samples by were 36.490–55.490, which supports our findings [33].

The metabolism of cholesterol, fat, and glucose is driven by Chromium and plays a very important role. Its deficiency creates high blood glucose, high body fat, and lower sperm count, and it is toxic and carcinogenic at high levels. Cr contamination mainly involves releases by the process of electroplating and disposal of waste containing Cr. Additionally, [27] reported that Cr concentration in soil was 20.83–104.83 mg $kg^{-1}$. Bekana et al. [40] also reported a 20.71–41.45 mg $kg^{-1}$ concentration of HMs in samples of soil. The Cd level in eggplant was highest in samples irrigated with sewage water, i.e., $7.6 \pm 0.17$, followed by canal water $5 \pm 0.26$ and tube well water $4.5 \pm 0.3$, respectively. The Cd in eggplant samples ranged from $3.1 \pm 0.35$–$4.5 \pm 0.3$ (tube well water) to $5.2 \pm 0.36$–$7.6 \pm 0.17$ (sewage water). These concentrations exceeded the allowable limits of EU 3 mg $kg^{-1}$ [17]. Cd is a HM as well, which is not essential. Even at low concentrations, it is highly poisonous. This creates disabilities in learning and hyperactivity (Hunt, 2003). It can be regarded as very toxic as a non-essential metal. In a similar study, [27] reported the range of Cd concentration (mg $kg^{-1}$) in soil samples as 2.82 to 4.77 mg $kg^{-1}$, [40] reported the same as 0.79–412.16 mg $kg^{-1}$. The FAO and WHO (2001) allowable limit of Cadmium in the soil is 3 mg $kg^{-1}$.

However, daily human and animal absorption within a certain range of levels (UP to 150 µg/day) is regarded to be crucial for carbohydrates and lipid metabolism [41]. In a similar study, [40] reported vegetables generally contained the highest concentrations of Cr (12–14 mg $kg^{-1}$) at Haramaya University vegetable farm when compared with the present study. In this research, the Cadmium contents in samples of vegetables irrigated by sewage, i.e., $4.8 \pm 0.36$, followed by canal water $1.1 \pm 0.4$ and tube well water $1 \pm 0.15$, respectively. The Cd in eggplant samples ranged from $0.3 \pm 0.20$–$1 \pm 0.4$ (tube well water) to $3.7 \pm 0.5$ to $4.8 \pm 0.36$ (sewage water). These concentrations were greater than the allowable limits of FAO and WHO 0.05 to 0.2 mg $kg^{-1}$.

For the outcomes obtained from this research compared with studies done by [27] the amount of Cd in vegetable samples was 1.2 to 2.5 mg $kg^{-1}$. In line with this result reported by Prabu, [42] found that, in leafy vegetables such as lettuce, Swiss chard, spinach, and radish, accumulation of Cadmium (Cd) was greater than in other vegetables (*Raphanus sativus*) when compared with the present study. Similarly, [39] indicated that Cd concentration in cultivated vegetables from domestic wastewater ranges from 0.14 mg $kg^{-1}$ spinach and 0.30 mg $kg^{-1}$ brinjal (*Solanum melongena*) based on the dry weight.

Additional sources of these elements for plants may include rainfall, atmospheric dust, plant protection agents, and fertilizers that can be absorbed through the leaf blades. The levels of HMs in vegetables in the study area of the present research work were found to be much greater than the allowable limits given by WHO [43]. Contamination of the food web by HMs has turned into a major problem in the last few years due to its prospective build-up through contaminated water, land, and atmosphere in bio systems [44].

They are usually considered to be the most damaging impacts of human health exposure, even at low levels, and they are heterogeneous, including but not restricted to neurotoxic and carcinogenic activities [45]. Metals that are toxins to plants when present in the soil at concentrations above tolerance, such as Fe, Zn, and Co, are observed to be health-threatening, and they are all known to cause significant health issues in response to environmental pollution [46].

For all plants and animals, Fe is by far the most important and abundant. At the other side, however, it triggers tissue harm and some other illnesses in human beings at elevated concentrations, Anemias and neurodegenerative circumstances in humans are due to Fe [47]. Tsafe et al. [48] recorded in the studied soils a 'value of 195.25 mg kg$^{-1}$. The Pb concentration in eggplant was highest in samples irrigated with sewage water, i.e., 348.9 $\pm$ 0.70 mg kg$^{-1}$, followed by canal water 320.4 $\pm$ 1.04 mg kg$^{-1}$ and tube well water 308$\pm$ 1.81 mg kg$^{-1}$, respectively. The Pb in eggplant samples ranged from 0.5 $\pm$ 0.18 mg kg$^{-1}$ (tube well water) to 6.2 $\pm$ 0.70 mg kg$^{-1}$ (sewage water). These concentrations were within the allowable limits of EU 300 mg kg$^{-1}$.

Lead has an estimated retention time of 150 to 5000 years and is one of the more constant metals [49]. It's a heavy metal(oid) that is not essential. Pb creates oxidative stress and, by disrupting the sensitive antioxidant equilibrium of mammalian cells, adds to the pathogenesis of Lead poisoning. In the body, high concentrations of Pb accumulation trigger disease, central nervous system pain, anemia, colic, and brain harm [50]. However, in soils from different dumpsites, the Pb concentration of 3500 to 6860 mg kg$^{-1}$, recorded very elevated concentrations of Lead [51]. High soil values of 1340 to 1693 mg kg$^{-1}$ have been also recorded by Aluko et al. [52].

Even at lesser concentrations, Pb has a dangerous impact on health. Exposure to Lead excess of 0.01 mg kg$^{-1}$ may cause health effects, as it can result in possible fetal neurological damage, abortion, and other issues in children under the age of 3 [48]. Nickel concentration in eggplant was highest in samples irrigated with sewage water, i.e., 86.4 $\pm$ 0.68, followed by canal water 82.8 $\pm$ 0.60 and tube well water 77.2 $\pm$ 0.35, respectively. The Ni in eggplant samples ranged from 75.9 $\pm$ 0.37 to 77.2 $\pm$ 0.35 (tube well water) to 81.5 $\pm$ 0.45 to 86.4 $\pm$ 0.68 (sewage water). These concentrations were above the allowable limits of EU, which is 75 mg kg$^{-1}$. The Ni concentrations found in soil samples by [31] were within the allowable limit of FAO and WHO.

Akubugwo et al. [51] investigated the Fe in vegetables, which was even greater than 147.41 mg kg$^{-1}$. Nearly all the samples have very elevated concentrations compared to all allowable concentrations of Fe. The elevated Fe and other HM levels found in the plant components can be due to the plant capacity to absorb HMs from polluted soils. Iron is a key component in any crop with many important biological functions, such as photosynthesis, chloroplast development, and biosynthesis of chlorophyll. Humans have an increased risk for several estrogen-induced cancers due to excessive build-up of Fe in human beings [53]. Lead (Pb) concentration in eggplant was highest in samples irrigated with sewage water, i.e., 6.5 $\pm$ 0.36 mg kg$^{-1}$, followed by canal water 1.2 $\pm$ 0.36 mg kg$^{-1}$ and tube well water 0.8 $\pm$ 0.3 mg kg$^{-1}$, respectively. The Pb in eggplant samples ranged from 0.2 $\pm$ 0.2 to 0.8 $\pm$ 0.3 (tube well water) to 3.9 $\pm$ 0.20 to 6.5 $\pm$ 0.36 mg kg$^{-1}$ (sewage water). These concentrations were greater than the allowable limits of WHO and FAO 425.5 mg kg$^{-1}$. Lead is a toxic component that can be damaging to crops, although crops generally demonstrate the capacity to accumulate immense amounts of Pb without noticeable modifications in yield. The Pb accumulation of 48, in many crops, may exceed the maximum permissible limit several hundred times. The airborne Pb build-up in leafy vegetables surpasses the portion of the soil transported by the roots. Airborne Lead is primarily piled up at the leaf layer and can be cleaned mainly by cleaning the vegetables [54].

The Fe concentrations found in tube well water samples by Rasool et al. (2016) were 173 mg L$^{-1}$, which supports our findings. Rasool et al. (2016) studied and found the highest level of Zn, i.e., 52 mg L$^{-1}$, in tube well water, which supports the finding of this study. The Cr concentrations found in tube well water samples by Rasool et al. (2016) were

12 mg L$^{-1}$, which supports our findings. The Cd concentrations found in tube well water samples by Rasool et al. (2014) were 4, which supports our findings. The Pb concentrations found in tube well water samples by [9] were 23 mg L$^{-1}$, which supports our findings.

The availability and transfer of HMs are dependent on different factors of soil, including organic carbon, pH, cation exchange capacity, mineral and clay contents, and oxidation-reduction [10]. Speciation and solubility of HMs in soil means soil pH has the most damaging role of all soil factors [55,56]. The HM solubility increases if the lower pH increases, hence increasing the plant accumulation and uptake and creating potential threats to the environment [57] and human health.

## 5. Conclusions

In this study, the trend of HM accumulation in soil and eggplant was measured for different irrigation water sources. The results showed that sewage water samples' turbidity was above the permissible limits of PEQs. The pH in tube well water was within the permissible range, while it was higher for sewage and canal water samples. The average TDS in canal water was followed by sewage water. The findings of HM accumulations in soil showed that Zn, Fe, Pb, Cr, Ni, and Cu showed relatively greater values for wastewater, canal, and ground water-irrigated lands. Overall the translocation factor of HMs in plant samples was, in order, Cadmium (Cd) > Zinc (Zn) > Nickel (Ni) > Iron (Fe) > Lead (Pb). On the basis of this study, it is concluded that irrigation source, especially sewage, is a major source of heavy metal(oid) contamination in eggplant. Further studies are required to investigate the gaps in implementation of policies for irrigation water quality.

**Author Contributions:** Conceptualization and software and supervision, N.E. and R.N.; methodology and field data collection, Y.T.; visualization and validation, W.A.K.; formal analysis, R.I.; data curation, S.A., U.R., E.A.M., I.U. and H.O.E.; formal analysis, S.A., E.A.M., I.U. and H.O.E.; funding acquisition, E.A.M., I.U. and H.O.E.; writing—review & editing, S.A., E.A.M., I.U. and H.O.E. All authors have read and agreed to the published version of the manuscript.

**Funding:** Researchers Supporting Project number (RSP2023R118), King Saud University.

**Data Availability Statement:** All the data available within first and second author as main supervisors of the study.

**Acknowledgments:** Our sincere appreciation to Researchers Supporting Project number (RSP2023R118), King Saud University, Riyadh, Saudi Arabia.

**Conflicts of Interest:** The authors declare no conflict of interest.

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
