# Peer review of "Assessment of Heavy Metal(oid)s Accumulation in Eggplant and Soil under Different Irrigation Systems"

_water, doi:10.3390/w15061049_

Round 1

Reviewer 1 Report

The reviewed manuscript is dealing with the assessment of Heavy Metal(oid)s accumulation in vegetables and soil under different irrigation systems. This is an interesting and valuable contribution, but the current form should be improved. Some tables are only mentioned without proper explanation and integration into the manuscript (Statistical analysis and Translocation Factor are present only as a table, without any comment). The manuscript's serious shortcoming is its lack of discussion of the data; no effort was made to provide an explanation of the results.

1. The abstract: it must be improved because it is inadequate. For Example, the authors gave a long space taking about sampling (Lines 29:33), turbidity, and TDS (Lines 34:39). More space should be given to bringing up the concentrations of HMs in water, soil, and plants, as this is the study's objective.

2. The Introduction: this section must be revised and rearranged. The authors presented a lot of valuable information on various topics, but it is overlapping and repetitive in a way that makes it difficult to track the main ideas in this section. For Example, the authors presented information about sources of HMs, the effect of HMs on plants and human health, and some information about Pakistan in different parts of this section. I recommend rearranging the introduction to present some key ideas, such as the significance of soil pollution, the sources of soil pollution with HMs, the demand for clean water, the effect of soil pollution on plants, the impact of HMs on plants and humans health, and any information about water and soil pollution in the study area. Furthermore, the last paragraph of the introduction should clearly explain the objective of the study.

3. Line 53; please remove Heavy Metals.

4. Line 75; please define ROS in its first appearance.

5. Line 81; there is no need to provide information about the abundance of elements in the earth's crust; instead, be specific and simply focus on sources and effects.

6. Line 94; “ground water” change to groundwater (here and onward for Figures 3 and 4).

7. Lines 138:140; are these your results?

8. Line 167; please double-check the references; APHA (1999) was left from the list.

9. Lines 192:198; the authors must report on Quality Control and Assurance, taking into account information such as analytical quality control, accuracy, certified reference materials, and detection limits.

10. Lines 193:194; please remove the ICARDA and APHA hyperlinks and add them to the references list.

11. Lines 196:197; please add PEQs, WHO, and FAO to the references list.

12. Table 1; please add EU to the references list.

13. Line 202; what is the value of this limit?

14. Figure 2; please improve the quality of this figure.

15. Table 2; this table is not mentioned in the text. Please specify which value is for correlation and which is for p-value. F1???

16. Lines 225:236; please rewrite this paragraph to avoid repeating information.

17. Lines 245:246; “HMs like Cd and Pb in general do not have a valuable effect on humans and no mechanism of homeostasis is known for these” this is not correct, please see https://doi.org/10.3390/toxics8040086; https://doi.org/10.1007/s40201-020-00455-2; https://doi.org/10.1371/journal.pone.0207423; https://doi.org/10.3389/fphar.2021.643972; https://doi.org/10.3390/toxics10090524; https://doi.org/10.3390/su132413538.

18. Lines 254:300; please rewrite this part to avoid repeating information.

19. Line 302; please comment on the data presented in Table 3.

20. The Discussion Section: the authors failed to discuss the presented results. This section must be completely revised and represented.

21. Line 305; where is Figure 5?

22. The Conclusion: should be revised and restructured based on the study's key findings.

Reviewer 2 Report

In this study, the authors evaluated the effects of three different irrigation water sources on the accumulation of heavy metals in eggplant and soil. However, paper needs very significant improvement before acceptance for publication. My detailed comments are as follows:

1.     The title of the manuscript does not mention eggplant. Eggplant can not represent all vegetables, suggest the author to add eggplant in the title. The title here cover too wide a range, suggest that the author change to a more accurate title.

2.     The introduction section is illogical and the overall idea is confusing. It is recommended that the author reorganise the introduction to focus on the need for irrigation water, the current situation of heavy metal pollution and the dangers of heavy metal pollution, and to combine the current situation of the country, rather than a separate paragraph as at present.

3.     In “3.3 Heavy Metal Concentration in Vegetable Samples” section, data figure or table supporting research results is missing.

4.     The author mentions “The Assessment of heavy metal (oid)s accumulation in vegetables and soil under different irrigation systems is summarized in figure 5.” on line 304-305. However, figure 5 is not found in this manuscript.

5.     There is no any research data in the manuscript to support the conclusion of “The level of HMs in vegetables varies by the ability of plants to selectively translocate from the contaminated soil to the plant's fruit (eggplant) and accumulate some of these elements.”

6.     Many abbreviations were not written with their full names when they first appeared in the manuscript, please check and add.

7.     I suggest that the authors redraw Figure 1 and use it as a model at the end of the paper.

Reviewer 3 Report

The research sought to investigate the effect of different sources of irrigation water on soil and irrigated crops in relation to the accumulation of heavy metals.

The experimental model is interesting and brings expressive and important results to seek to resolve an important public health issue in the region. However, important issues should be pointed out in order to better clarify some aspects of the manuscript. 1 – Firstly, it is important to clarify throughout the work whether the sewage water used is treated or untreated. Is this pure sewage? How can this sewer be described? Is it only residential? 2 – The introduction seems a little confusing towards the end, as in addition to not describing the objective of the research, it presents a sentence that seems to be a result of the research carried out. Revise and rewrite the end of the introduction, correcting these elements. 3 – In the methods section, figure 1 is very important and was very well elaborated. However, for a better understanding of the method, the authors should describe the process of digestion of the samples and the process of reading in spectrophotometry. 4 – In the results section, the high values ​​of metals in rural sewage water (SW2) and also in rural wells (TW2) are very interesting. In general rural sewage water should be less contaminated. In my opinion, based on these data, it is important that the authors make a small geological description of the region, as this can better inform the origin of the metals. 5 – In my opinion, Figure 5 was missing, which could show the amount of metals in the vegetable samples. This figure is fundamental to show, like the others, what were the differences observed in the vegetable. 6 – It was not clear to me if the plants were irrigated with the tested waters. If so, the irrigation process, time, flow, type of irrigation must be presented.

At the end of the results, a table appears showing the Translocation Factor (TF) from the soil to the plant. Then comes the question: did the research analyze the transfer from water to the plant or from the soil to the plant? In both cases, it is necessary to describe the irrigation time, the irrigation process, the phase of the irrigated plant and how long this whole procedure lasted to better understand if there was a direct transfer from water or if the transfer really was via the soil.

Reviewer 4 Report

Please see in the attach.

Round 2

Reviewer 1 Report

The authors worked hard to improve their manuscript. However, some comments have not been addressed, and some errors need to be corrected. This round, I tried to be more specific in order to avoid any confusion. In general, it is an interesting study. However, the authors' findings were not presented clearly. Readers must work hard to understand what the authors are trying to say. This will have a negative impact on the research's readability and citation rate. I am sure that the authors will do their best to improve the research to the greatest standard without the need for another round of review.

1. Lines 31:36; This paragraph is quite long. Please keep it brief and to the point (to reduce unnecessary abbreviations in the abstract).

2. The Introduction: this section must be revised and rearranged. The authors presented a lot of valuable information on various topics, but it is overlapping and repetitive in a way that makes it difficult to track the main ideas in this section. For Example, the authors presented information about sources of HMs, the effect of HMs on plants and human health, and some information about Pakistan in different parts of this section. I recommend rearranging the introduction to present some key ideas, such as the significance of soil pollution, the sources of soil pollution with HMs, demand for clean water, the effect of soil pollution on plants, the impact of HMs on plants and human health, and any information about water and soil pollution in the study area. Furthermore, the last paragraph of the introduction should clearly explain the objective of the study.

 Ans: Information is rearranged and errors are removed (Page #3&4)

 Reviewer: There was no noticeable rearrangement??

 3. Figure 2; please improve the quality of this figure and insert the figure caption.

4. Table 2; this table is not mentioned in the text. Please specify which value is for correlation and which is for p-value. Of course, the authors know which value represents correlation and which represents p-value, but the reader who is not a specialist may not be able to distinguish between the two. For example TDS (0.058 – 0.767).

5. Lines 233:244; please rewrite this paragraph to avoid repeating information. It is preferable to only mention each element once. Why was the order of the elements mentioned with average values first, and then separately with range values?

6. Line 246; Please improve the quality of this figure and revise Figure No (Figure 3).

7.  Lines 253:254; “HMs like Cd and Pb in general do not have a valuable effect on humans and no mechanism of homeostasis is known for these” this is not correct, please see https://doi.org/10.3390/toxics8040086; https://doi.org/10.1007/s40201-020-00455-2; https://doi.org/10.1371/journal.pone.0207423; https://doi.org/10.3389/fphar.2021.643972; https://doi.org/10.3390/toxics10090524; https://doi.org/10.3390/su132413538.

 Ans: references added

Reviewer: I didn't ask the authors to include any references at this point. I referred to this article for the author's information about the incorrect sentence regarding the health effects of Cd and Pb.

8. Line 257; Please improve the quality of this figure and revise Figure No (Figure 4).

9. Line 283; Please improve the quality of this figure and revise Figure No (Figure 5).

Author Response

Reviewer-1

The authors worked hard to improve their manuscript. However, some comments have not been addressed, and some errors need to be corrected. This round, I tried to be more specific in order to avoid any confusion. In general, it is an interesting study. However, the authors' findings were not presented clearly. Readers must work hard to understand what the authors are trying to say. This will have a negative impact on the research's readability and citation rate. I am sure that the authors will do their best to improve the research to the greatest standard without the need for another round of review.

  1. Lines 31:36;This paragraph is quite long. Please keep it brief and to the point (to reduce unnecessary abbreviations in the abstract).

Thank you for your comment regarding the length of the paragraph. We will certainly review and revise the paragraph to ensure that it is brief and to the point while still conveying the necessary information. We appreciate your feedback and will make every effort to ensure that the abstract is clear, concise, and effectively communicates the main points of the research.

  1. The Introduction:this section must be revised and rearranged. The authors presented a lot of valuable information on various topics, but it is overlapping and repetitive in a way that makes it difficult to track the main ideas in this section. For Example, the authors presented information about sources of HMs, the effect of HMs on plants and human health, and some information about Pakistan in different parts of this section. I recommend rearranging the introduction to present some key ideas, such as the significance of soil pollution, the sources of soil pollution with HMs, demand for clean water, the effect of soil pollution on plants, the impact of HMs on plants and human health, and any information about water and soil pollution in the study area. Furthermore, the last paragraph of the introduction should clearly explain the objective of the study.

 Ans: Information is rearranged and errors are removed (Page #3&4)

 Reviewer: There was no noticeable rearrangement??

 Ans: Information is rearranged and changes are highlighted

  1. Figure 2;please improve the quality of this figure and insert the figure caption.

Figure replaced and caption added

  1. Table 2;this table is not mentioned in the text. Please specify which value is for correlation and which is for p- Of course, the authors know which value represents correlation and which represents p-value, but the reader who is not a specialist may not be able to distinguish between the two. For example TDS (0.058 – 0.767).

Ans: Comment incorporated

  1. Lines 233:244; please rewrite this paragraph to avoid repeating information. It is preferable to only mention each element once. Why was the order of the elements mentioned with average values first, and then separately with range values?
  2. Line 246; Please improve the quality of this figure and revise Figure No (Figure 3).

Ans: The quality of figure improved and all the figure numbers correct accordingly in text as well as figure caption

  1. Lines 253:254; “HMs like Cd and Pb in general do not have a valuable effect on humans and no mechanism of homeostasis is known for these” this is not correct, please see https://doi.org/10.3390/toxics8040086; https://doi.org/10.1007/s40201-020-00455-2; https://doi.org/10.1371/journal.pone.0207423; https://doi.org/10.3389/fphar.2021.643972; https://doi.org/10.3390/toxics10090524; https://doi.org/10.3390/su132413538.

Corrected

 Ans: references added

Reviewer: I didn't ask the authors to include any references at this point. I referred to this article for the author's information about the incorrect sentence regarding the health effects of Cd and Pb.

  1. Line 257; Please improve the quality of this figure and revise Figure No (Figure 4).

 Ans: The quality of figure improved and all the figure numbers correct accordingly in text as well as figure caption

  1. Line 283; Please improve the quality of this figure and revise Figure No (Figure 5).

 Ans: The quality of figure improved and all the figure numbers correct accordingly in text as well as figure caption

Reviewer 2 Report

There are still some issues:

1. Line 78 Reactive Oxidative species, R and O should be lowercase letters;

2.  Line 125 “Mercury”,  M should be lowercase letters;

3. Line 206, something were missing?

4. There are two unit formats in the text and the figure (Fig. 4), like mg L-1? mg/kg?

5. “mgkg-1” space should  be included.

6. Line 298: “5.5±0.20-6.1 ± 0.25 mg L-1”, it is confusing.

There a lot of similar mistakes, here, I just give some example. It think the manuscript should be checked by academic expert before accepting for publication.

Author Response

Reviewer-2

There are still some issues:

  1. Line 78 “Reactive Oxidative species”, R and O should be lowercase letters;

Ans: Corrected

  1. Line 125 “Mercury”,  M should be lowercase letters;

Ans: Corrected

  1. Line 206, something were missing?

Ans: corrected

  1. There are two unit formats in the text and the figure (Fig. 4), like mg L-1? mg/kg?

Ans:Corrected

  1. “mgkg-1” space should  be included.

Ans: corrected

  1. Line 298: “5.5±0.20-6.1 ± 0.25 mg L-1”, it is confusing.

There a lot of similar mistakes, here, I just give some example. It think the manuscript should be checked by academic expert before accepting for publication.

Ans: corrected

Reviewer 3 Report

Ícone "Verificada pela comunidade" The changes made improved the text and in my opinion left the manuscript in conditions to be published

Author Response

(The authors gave the same response as above.)

Reviewer 4 Report

I recommend the manuscript for publication.

Author Response

(The authors gave the same response as above.)
